# Picrotoxin Delineates Different Transport Configurations for Malate and γ Aminobutyric Acid through TaALMT1

**DOI:** 10.3390/biology11081162

**Published:** 2022-08-02

**Authors:** Sunita A. Ramesh, Yu Long, Abolfazl Dashtbani-Roozbehani, Matthew Gilliham, Melissa H. Brown, Stephen D. Tyerman

**Affiliations:** 1Biological Sciences, College of Science and Engineering, Flinders University, Bedford Park, SA 5042, Australia; abolfazl.dashtbaniroozbehani@flinders.edu.au (A.D.-R.); melissa.brown@flinders.edu.au (M.H.B.); 2ARC Centre of Excellence in Plant Energy Biology, Department of Plant Science, School of Agriculture, Food and Wine, Waite Research Institute, The University of Adelaide, Waite Campus, Glen Osmond, SA 5064, Australia; yu.long@henu.edu.cn (Y.L.); matthew.gilliham@adelaide.edu.au (M.G.); steve.tyerman@adelaide.edu.au (S.D.T.); 3State Key Laboratory of Crop Stress Adaptation and Improvement, School of Life Sciences, Henan University, Kaifeng 475001, China

**Keywords:** GABA, ALMTs, signalling, picrotoxin, binding site

## Abstract

**Simple Summary:**

Gamma aminobutyric acid (GABA) a non-protein amino acid is an archaic signalling molecule for cellular communication across all kingdoms of life with distinct signalling roles in animals. Emerging research establishes GABA as a signalling molecule *in planta*. GABA modulates numerous developmental processes and negatively regulates root growth, stomatal aperture, and anion flux under abiotic stress. A putative GABA binding site with homology to the mammalian GABA binding site has been identified in a family of proteins (Aluminium Activated Malate Transporters -ALMT) involved in mediating anion flux and conferring plant aluminium tolerance. Studies have shown that ALMTs are regulated by GABA and transport both anions and GABA. To gain further insights into the mechanism of GABA regulation, it is essential to differentiate between the two transport modes. Pharmacological agents used to characterise mammalian GABA receptors have proved useful in gaining insights into GABA regulation of plant processes. In this study, picrotoxin an inhibitor of anion flux in mammalian GABA receptors was used to investigate the pathways for anion and GABA transport in ALMTs. Results suggest that picrotoxin inhibits anion flux but not GABA flux and can be used to gain further insights into mechanism of GABA regulation of plant proteins.

**Abstract:**

Plant-derived pharmacological agents have been used extensively to dissect the structure–function relationships of mammalian GABA receptors and ion channels. Picrotoxin is a non-competitive antagonist of mammalian GABA_A_ receptors. Here, we report that picrotoxin inhibits the anion (malate) efflux mediated by wheat (*Triticum aestivum*) ALMT1 but has no effect on GABA transport. The EC_50_ for inhibition was 0.14 nM and 0.18 nM when the ALMTs were expressed in tobacco BY2 cells and in *Xenopus* oocytes, respectively. Patch clamping of the oocyte plasma membrane expressing wheat ALMT1 showed that picrotoxin inhibited malate currents from both sides of the membrane. These results demonstrate that picrotoxin inhibits anion efflux effectively and can be used as a new inhibitor to study the ion fluxes mediated by ALMT proteins that allow either GABA or anion transport.

## 1. Introduction

Emerging evidence compiled over the last two decades suggests that gamma-aminobutyric acid (GABA) is an endogenous signalling molecule in plants [1,2,3,4,5,6,7,8,9,10]. In response to both abiotic (acidosis, hypoxia, salinity, drought, and cold) and biotic (pathogens, viruses, and herbivory) stress, GABA concentrations can change rapidly in plants [1,4,11,12,13,14,15,16,17,18]. Micromolar concentrations of GABA (and its analog muscimol) regulate anion currents through the ALMT family of proteins from various plant species [4], and the GABA-induced inhibition of anion transport via an ALMT protein localised to the tonoplast in guard cells can regulate stomatal aperture under drought stress [19]. This modulation of anion flux facilitated by ALMTs presents a probable process by which GABA can function as a stress signal in plants [4], depend on the amino acids present within a putative GABA binding motif in ALMTs, and share homology with the GABA binding site of mammalian GABA_A_ receptors. The site-directed mutagenesis of particular amino acids in the putative GABA binding motif alters the GABA sensitivity of plants [4], and the patch clamping of the wheat ALMT1 (*TaALMT1*) that is expressed in *Xenopus* oocytes indicates that GABA is inhibitory on the cytoplasmic side of the protein [20].

It is possible to manipulate the internal GABA concentrations in plant cells using the pharmacological agents (e.g., amino-oxyacetate (AOA) and vigabatrin) that have been used to characterise mammalian GABA receptors [21,22,23,24,25] or according to the mutagenesis of the enzymes involved in GABA synthesis [26,27]. The manipulation of endogenous GABA concentrations using pharmacological agents such as AOA and vigabatrin have provided insights into the association between anion flux and GABA modulation in anion channels, although these agents are also directly manipulated *TaALMT1* activity [28]. Malate efflux via ALMTs is negatively correlated with internal GABA concentrations, and in ALMT with mutations in the putative GABA binding site, this relationship is eliminated [28]. In *Xenopus laevis* oocytes, tobacco BY2 cells, and yeast, *TaALMT1* expression decreases endogenous GABA concentrations by facilitating GABA efflux from the cells, and that GABA can also be transported into the cell via ALMTs at low pH levels [28]. The capacity for GABA efflux via ALMTs seems to be higher when compared to malate efflux on a molar basis, and this is expected to have wide-ranging impacts on the carbon/nitrogen balance and signalling in plants.

ALMTs in general, and wheat roots in particular, efflux malate (anions) and GABA via ALMTs upon exposure to aluminium (Al) at low pH levels [29] and in response to external anions at neutral and high pH levels [4,30,31]. The mechanism for the transport and efflux of GABA across cell membranes and its interaction with malate through ALMT channels is suggestive of these proteins having a dual function [28]. However, the question that arises is whether both malate and GABA are transported through the same pores in ALMT or via a different pathway in the same protein simultaneously. The recently described cryogenic electron microscopy (cryo-EM) structure of *Arabidopsis thaliana* AtALMT1 reveals that the ALMT1 dimers assemble as an anion channel and that trivalent aluminium ions (Al^3+^) bind on the extracellular side, leading to conformational changes in the transmembrane (TM) 1–2 loop and in the TM5-6 loop that result in the opening of an extracellular gate for malate flux [32]. One permeation pathway for ion flux is predicted with residues D49, E156, and D160 that are important for Al^3+^ activation. However, the pathway for GABA flux through the protein or binding was not examined for AtALMT1. If a single permeation pathway exists through ALMT1, the protein may have two conformational states that affect selectivity: one that is permeable to malate, and one that is permeable to GABA, with the conformation of the pore depending on whether it is GABA or malate that is binding to the protein. The apoplastic and cytoplasmic concentrations of both malate and GABA may be finely titrated, and each regulates and is regulated by the other in the cells.

In animal studies, plant-derived pharmacological agents have been used extensively to dissect the structure and function of GABA receptors and ion channels. Picrotoxin derived from the seeds of *Anamirta cocculus* can act on GABA_A_ receptors as either a non-competitive channel blocker (Cl^−^ channels) [33,34] or by mediating inhibition via an allosteric mechanism [35,36,37]. Interestingly, the picrotoxin-mediated inhibition of glycine receptors (GlyR) is competitive [38]. Picrotoxin is not structurally similar to GABA, but it inhibits ion flow via the chloride channels activated by GABA by binding within the ion channel rather than at GABA recognition sites [39]. Picrotoxin may act by preferentially binding to an agonist-bound form of the receptor and reducing the frequency of channel openings [34]. Similarities between the GABA binding region in mammalian GABA_A_ receptors and the putative GABA motif in the ALMT family of anion channels have enabled the use pharmacological agents to dissect and characterise the function of GABA binding site in plants.

In this study, picrotoxin was tested on a near-isogenic line (NIL) of wheat (ET8) that is tolerant to aluminium at low pH levels [40] as well as on *TaALMT1* and ALMT from *Vitis vinifera* (*VvALMT9*) expressed in tobacco BY2 cells and *Xenopus* oocytes, respectively, to assess if it inhibited anion and GABA efflux. The exogenous application of picrotoxin had an inhibitory effect on malate efflux; however, there was no effect on the efflux of GABA mediated by the wheat and grapevine ALMTs. Interestingly, similar results were obtained in the site-directed mutant of *TaALMT1*, i.e., *TaALMT^F213C^*, which is impaired during the GABA-induced regulation of malate transport [4,28]. Continued GABA efflux mediated by ALMTs upon exposure to picrotoxin suggests that ALMTs may exist in two conformational states: one that is permeable to malate and inhibited by picrotoxin, and one that is permeable to GABA and insensitive to picrotoxin.

## 2. Materials and Methods

### 2.1. Chemicals

All the chemicals used in this study were obtained from Sigma-Aldrich (St. Louis, MO, USA). Picrotoxin (Pic) was dissolved in DMSO.

### 2.2. cRNA Synthesis

The mini Prep kit from Sigma-Aldrich was used to extract plasmid DNA from *TaALMT1*, the site-directed mutant *TaALMT1^F213C^*, and *VvALMT9* [4]. The restriction enzyme NheI was used to linearise 1 µg of plasmid DNA from each, and a mMESSAGE mMachine T7 Kit (Ambion, Austin, TX, USA) was used to synthesise the capped cRNA as per the manufacturer’s instructions.

### 2.3. Voltage-Clamp Electrophysiology

The two-electrode voltage clamp (TEVC) technique was performed on *Xenopus laevis* oocytes 2 d after injection with water/cRNA [4,41]. The oocytes were injected with 46 nL of RNase-free water using a micro-injector (Nanoject II, automatic nanoliter injector; Drummond Scientific) ± 16 to 32 ng cRNA (*TaALMT1* or *TaALMT1^F213C^*). Sodium malate (10 mM, pH 7.5) was injected into the oocytes 1 to 2 h before measurement. The low-pH experiments were performed in an ND88 solution with a pH of 4.5 ± aluminium chloride (100 μM AlCl_3_) ± picrotoxin (0.1 nM to 100 μM). The basal external solution for anion activation contained 0.7 mM CaCl_2_ and mannitol to 220 mOsm kg^− 1^ ± 10 mM sodium sulphate and picrotoxin (0.1 nM to 100 μM) buffered with 5 mM Bis-Tris propane and 2-(N-morpholino) ethane sulfonic acid (BTP/MES) to pH of 7.5 [40,42]. For the TEVC experiments, all of the experimental solutions were applied to both the gene-injected and control oocytes in the same order. Oocytes were selected arbitrarily, and experiments were carried out by alternating between the gene-injected and control oocytes to remove any bias. All the experiments conducted with *X. laevis* were approved by the University of Adelaide ethics committee.

### 2.4. Patch Clamping

Patch clamp electrophysiology was carried out as detailed in Long et al. (2020) [20]. An Axon 200B amplifier Axon Clampex 11 was used to acquire data via Digidata 1550B (Molecular Devices) [20]. As the number of channels in a patch was generally greater than one, it was not possible to obtain precise single-channel open-state probability (P_open_). Thus, NP_open_ analysis was conducted using Clampfit 11 software to calculate P_open_ over a specific time interval (usually 90 s) for each treatment. The total open times of all the channel events were divided by the total observation time. *N* is the number of channels in the patch estimated from the maximum number of simultaneous openings as Σ*N*_*j*=1_*t_j_j*/*TN*, where *t_j_* is the time spent with *j* = 1, 2, …, *N* channels open [43]. All of the membrane patches were isolated as per the protocol outlined by Maksaev and Haswell [44]. The details of the solutions used in the bath and pipette are described in the relevant figure legend. All the junction potentials between the pipette and bath solutions were measured as ~2.5 mV and were not adjusted.

### 2.5. Malate Efflux Measurements

*Triticum aestivum* ET8 seeds [39] were surface sterilised with bleach (1%), washed in water (3 times), and germinated on moist filter paper in the dark. The roots of the 3-day-old seedlings were exposed to 3 mM CaCl_2_ and 5 mM MES/BTP at either pH 4.5 or 7.5 ± treatments for 22 h in microfuge tubes. Malate flux from the roots was measured using a K-LMALR/K-LMALL kit (Megazyme) in an OMEGA plate-reading spectrophotometer (BMG). In brief, a 100 µL aliquot of treatment solution collected from the seedling roots was added to the contents of the kit, and the changes in the absorbance were measured as per the manufacturer’s instructions at 340 nm and were used to calculate the malate concentrations.

### 2.6. GABA Efflux Meaurements

Root tips (5 mm, 3–4 seedlings per replicate) were excised from the seedlings exposed to different treatments (22 h) and were frozen in liquid nitrogen. The frozen root tips were ground in liquid nitrogen, and a known weight (10–15 mg) of ground tissue was used for further extractions. In brief, the tissue was added to methanol, incubated at 25 °C for 10 min, and dried using a speedy vac. The pellets were washed in 70 mM lanthanum chloride (LaCl_3_) followed by precipitation using 1N potassium hydroxide (KOH). The supernatants were harvested and frozen until they were needed for further use. GABA concentrations were measured on an OMEGA plate-reading spectrophotometer following the GABase enzyme assay [4]. To determine the GABA efflux from oocytes, the oocytes that had been injected with *TaALMT1*, *VvALMT9* cRNA, or water (controls) were imaged in groups of four to five using a stereo zoom microscope (SMZ800) with a Nikon (cDSS230) camera 48 h after injection. The oocytes were exposed to treatment solutions (10 min), as indicated in the figure legends, followed by harvesting of the treatment solutions and snap freezing in liquid nitrogen. All the samples were assayed for GABA efflux using the GABase enzyme assay as per published protocols [28].

### 2.7. Tobacco BY2 Malate Efflux and GABA Efflux

*TaALMT1*, the site-directed mutant *TaALMT1^F213C^*, and empty vector constructs were transformed into tobacco (*Nicotiana tabacum*) BY2 suspension cells [4] and were cultured in MS medium supplemented with hygromycin (30 μg/mL) and 2,4-Dichlorophenoxyacetic acid (2,4-D; auxin; 3 mM) on a rotary shaker (∼100 rpm)for 48 h. The cell suspensions were centrifuged (5000× *g*) and washed in media containing 3 mM CaCl_2_, 3 mM sucrose, and 5 mM MES/BTP buffered to pH 4.5 or 7.5. Cell pellets (~0.15 g) were added to 50 mL tubes and exposed to 3 mL of solution containing 3 mM CaCl_2_, 3 mM sucrose, and 5 mM MES/BTP plus or minus the treatments, as indicated in the legends. The tubes were placed on a rotary shaker for 22 h followed by centrifugation and supernatant harvesting. The supernatants were subsequently analysed for malate and GABA efflux, as described above.

### 2.8. Logo and Homology Modelling

The Geneious Prime (2020) software application from Biomatters was used to create the sequence alignment between TaALMT1, human GABAR, and rat GABAR proteins and to generate the logo. A homology model for the wheat ALMT1 based on *Arabidopsis* AtALMT1 was generated as a homodimer using the SWISS-MODEL web server (http://swissmodel.expasy.org/, accessed 10 April 2022). To identify the putative binding sites of TaALMT1 for picrotoxin and the associated molecular interactions, molecular docking was performed with AutoDock Vina V.1.1.2 (http://vina.scripps.edu; accessed 10 April 2022) [45].

### 2.9. Statistics

Data analysis was carried out using GraphPad Prism software (version 9.02). Data are presented as the mean ± SE, and the number of replicates is shown in the figure legends. One-way and two-way ANOVA with post tests were performed during the statistical analysis to determine significance between individual treatments.

## 3. Results

### 3.1. Wheat Root Malate and GABA Efflux

The exogenous application of Al^3+^ at low pH levels was shown to stimulate malate and GABA efflux from the roots of wheat seedlings as well as to reduce the internal GABA concentration [GABA]_i_ [28]. When exposed to Al^3+^ (100 μM), the NIL ET8 (Al^3+^ tolerant) from wheat induced malate efflux from the roots of 3-day-old seedlings at pH 4.5 (Figure 1A). When the roots were treated with picrotoxin (10 μM) at pH 4.5 in the presence of Al^3+^ (100 μM), it completely inhibited the Al^3+^-stimulated malate efflux to that observed under basal (pH 4.5) conditions (Figure 1A). Measurements of the exogenous GABA demonstrated that the wheat roots had significantly higher GABA efflux when exposed to not only Al^3+^ but also to Al^3+^ + picrotoxin when compared to low pH conditions or picrotoxin exposure alone (Figure 1B).

The roots of ET8 seedlings were treated with a basal solution with pH 7.5 ± 10 mM Na_2_SO_4_ (to add an activating external anion) ± picrotoxin (10 μM). In the presence of external anions, the stimulated malate efflux (Figure 1C) was significantly inhibited by picrotoxin (Figure 1C). The corresponding GABA efflux was significantly higher in the presence of external anions both with and without picrotoxin (Figure 1D). The dimethyl sulfoxide (DMSO) that was used to dissolve the picrotoxin had no effect on malate or GABA efflux (Appendix A).

### 3.2. Malate and GABA Efflux in Xenopus Laevis Oocytes Expressing ALMT

Malate efflux was significantly higher in both the *TaALMT1-* and *VvALMT9*-expressing oocytes when they were exposed to Al^3+^ (100 μM, pH 4.5) or Na_2_SO_4_ (10 mM, pH 7.5), respectively (Figure 2A,C). Picrotoxin (10 μM) ± Al^3+^ (100 μM) / Na_2_SO_4_ (10 mM) did not inhibit GABA efflux from the oocytes (Figure 2B,D). The DMSO that was used to dissolve the picrotoxin had no effect on malate or GABA efflux (data not shown).

### 3.3. Malate and GABA Efflux in Tobacco BY2 Expressing TaALMT1 and TaALMT1^F213C^

Tobacco BY2 suspension cells expressing *TaALMT1* or the mutant protein (*TaALMT**^F213C^**)* were used to test whether the malate efflux stimulated by the different treatments was blocked by picrotoxin when compared to the empty vector-expressing cells (Figure 3: control). The addition of external SO_4_^2−^ (10 mM) to the medium stimulated malate efflux from the cells expressing both *TaALMT1* and the mutant protein *TaALMT1^F213C^* (Figure 3A). The addition of varying concentrations of picrotoxin (0.1 nm to 100 μM) in the presence of SO_4_^2−^ (10 mM) showed that malate efflux was inhibited in the presence of 0.1 nM picrotoxin in both the *TaALMT1-* and *TaALMT1^F213C^*-expressing cells (Figure 3A). Similar to the results observed in the NIL ET8 from wheat and in the oocytes expressing *TaALMT1*, GABA efflux was not blocked by picrotoxin for either the *TaALMT1-* or *TaALMT1^F213C^-*expressing cells (Figure 3B). These experiments were also carried out at pH 4.5 in the tobacco BY2 cells (data not shown), and the results were similar to those obtained at pH 7.5.

### 3.4. Picrotoxin Inhibits Malate Efflux with High Affinity

Picrotoxin blocks malate efflux mediated by ALMT in the roots of wheat seedlings and tobacco BY2 cells (Figure 3A). The dose response of malate efflux inhibition mediated by picrotoxin was examined in tobacco BY2 cells and by malate currents when using TEVC in *Xenopus* oocytes. The apparent EC_50_ for malate efflux inhibition by picrotoxin was 0.14 nM for tobacco BY2 cells expressing *TaALMT1* and 0.34 nM for cells expressing the *TaALMT1^F213C^* mutant protein (Figure 4A,B). In the oocytes expressing *TaALMT1*, the apparent EC_50_ for the inhibition of the malate-stimulated currents by picrotoxin was 0.18 nM (Figure 4C). Picrotoxin had no effect on the control oocytes (Appendix A).

### 3.5. Blockade of TaALMT1 Single Channels by Picrotoxin

*TaALMT1* ion channel events can be identified in the outside-out recordings on the plasma membrane of *TaALMT1*-expressing *Xenopus* oocytes via the patch clamp technique [20]. These channels show small single-channel openings at negative membrane voltages. The single-channel properties of *TaALMT1* on the outside-out membrane patches from oocytes were recorded when picrotoxin was present or absent in either the bath or the pipette, allowing for the assessment of how picrotoxin may access *TaALMT1*. Picrotoxin (1 μM) blocked malate-induced ion currents on both sides of the membrane, mainly by a reducing the channel open probability (P_open_) (Figure 5A), but there was also a significant reduction in the single-channel current when picrotoxin was present on the cytosolic side of the membrane (Figure 5B).

### 3.6. Probable Binding Site for Picrotoxin on ALMT

In mammalian studies, two amino acids in the α1 subunit of the GABA_A_R, valine (V257) and threonine (T261), have been shown to line the channel near the putative cytoplasmic end of the M2 membrane-spanning segment [35]. The sequence alignment of the M2 region of human and rat GABA_A_ receptors with TaALMT1 shows a putative motif in TM5 with the amino acids aspartate (D) and threonine (T), which can bind picrotoxin and may represent one of the regions that line the channel pores (Figure 6A). A recent cryo-EM structure of Arabidopsis ALMT1 identified residues that may be important for Al^3+^ activation and malate binding [32]. Based on this structure, a homology model for the wheat ALMT1 was generated (see methods). This model indicates that picrotoxin (shown in cyan) may bind in the TM5 region of the protein and may interact with the amino acids alanine (A163), aspartate, (E166) and tyrosine (Y167) (Figure 6B,C). Further work is required to confirm whether these residues are involved in picrotoxin binding and affect malate/GABA permeation through TaALMT1.

## 4. Discussion

Picrotoxin has been shown to inhibit most anionic cysteine-loop ligand-gated ion channels, including homomeric GlyR [38], the GABA type-A receptor (GABA_A_R) [34,46], and the invertebrate glutamate receptor chloride channel (GluClR) [47].

In this study, the malate anion efflux mediated by *TaALMT1* was inhibited by picrotoxin in wheat roots, in tobacco BY2 cells, and in *X. laevis* oocytes (Figure 1, Figure 2, and Figure 4). This may be a general feature of ALMTs since the malate-induced inward currents from the oocytes expressing *VvALMT9* were also inhibited by picrotoxin. These results suggest that picrotoxin might bind to the ALMT protein, causing a conformational change leading to channel inactivation, or may interact with a malate binding site, leading to the inhibition of anion flux. In mammalian studies, evidence suggests that picrotoxin interaction occurs with the residues lining the central ion pore in the second transmembrane region (M2) of the five subunits of GABA receptors to inhibit chloride flux [36]. The mutation of a threonine to phenylalanine in any one of the subunits is sufficient to impart resistance to picrotoxin [36]. Although these results imply that picrotoxin binds in the pore, it is unlikely that the mechanism of picrotoxin inhibition is caused by pore occlusion. Studies on the human GlyR α1 subunit have demonstrated that picrotoxin can be trapped in the pore when the gate closes [48]. Thus, it is also possible that picrotoxin is trapped in the ALMT channel, affecting its opening (Figure 5), and leading to the inhibition of anion efflux. A significant reduction in the open-channel current also suggests binding within the pore, but only when picrotoxin is present on the cytoplasmic side. This is similar to the greater GABA efficacy observed during the blockade of *TaALMT1* malate currents from the cytoplasmic side [20], but picrotoxin differs from GABA block, which only shows a reduction in the ion channels P_open_, by also showing a reduction in the single-channel currents. It should be noted that GABA is neutral at the pH level used in patch clamping, so it does not resolve as ion currents when transported via *TaALMT1*.

ALMT proteins can facilitate both anions (malate) and GABA fluxes [28]. In the current study, picrotoxin inhibited anion flux from both *TaALMT1* (wildtype) and its site directed mutant *TaALMT1^F213C^,* impaired in GABA sensitivity. However, in wheat roots as well as in heterologous expression systems, picrotoxin did not inhibit the GABA efflux facilitated by ALMT expression (Figure 1 and Figure 2). The DMSO used to dissolve the picrotoxin had no effect on the either the malate or GABA fluxes mediated by the ALMT proteins. These results suggest that picrotoxin does not bind to the GABA binding site/s in the protein and that any conformational change that occurs in the ALMT proteins due to picrotoxin binding does not affect GABA transport. In mammalian studies, it is well established that agonist binding leads to a conformational change in the extracellular ligand binding domain, and this is transmitted via conformational changes in the M2–M3 loop of GABA_A_R to the channel activation gate [49].

Picrotoxin has been suggested to allosterically inhibit the ligand-gated ion channels in various studies, including in single-channel kinetic and conductance analysis studies [34,50], studies on the interactions between picrotoxin analogs [51], site-directed mutagenesis studies [38,52] and electrophysiology and fluorescence labelling studies [53]. The replacement of the valine and threonine residues in the M2 region with cysteine and the co-application of picrotoxin with GABA and sulfhydryl reagents has demonstrated that the blocked state of the channel is different from the closed state [35]. In this study, if picrotoxin acted as an open-pore blocker, then it would be expected that both the anion and GABA efflux mediated by the ALMT proteins would be inhibited. However, the lack of GABA efflux inhibition caused by picrotoxin indicates that this agent could function via an allosteric mechanism. The TaALMT1 from wheat shares 63% amino acid similarity to the *Arabidopsis* AtALMT1, and it is thus highly probable that the malate flux mediated by TaALMT1 employs a similar mechanism to that identified in AtALMT1. Homology modelling of TaALMT1 shows that picrotoxin-binding residues project into the pore (Figure 6B,C), strengthening the probability that it acts on ion fluxes via an allosteric mechanism. However, site-directed mutagenesis and characterisation need to be carried out to confirm the role of these residues in binding picrotoxin and if these residues are responsible for the allosteric inhibition of anion flux similar to that observed mammalian GABA_A_R.

Our current model of GABA and malate transport via TaALMT1 is summarised in Figure 7 and incorporates previous evidence [28]. In influx mode, GABA transport (1) into the cell can occur through GABA transporters such as GAT1 [54]. Malate efflux (2) across the membrane is mediated by the ALMTs (e.g., TaALMT1). In efflux mode, the GABA that is transported into the cell can bind to the GABA binding site (3) on TaALMT1 and block malate transport. We hypothesise that the GABA-induced blockage of malate transport, which is evident from the cytoplasmic side [20], is indicative of GABA inducing the TaALMT1 protein into a different conformation that prevents malate transport but allows GABA transport (4). In this conformation, it is likely that the picrotoxin binding site is unavailable. In the absence of GABA binding to its site on TaALMT1, picrotoxin can bind to its site and prevent malate efflux (5) from the cytoplasm. Thus, both malate and GABA efflux possibly occur via the same pathway through the protein, albeit in a different conformation, perhaps one that is related to the charge distribution at the selectivity filter, given that malate and GABA, although structurally similar, differ in that they are negatively charged and uncharged, respectively.

Taken together, picrotoxin could possibly inhibit TaALMT1 by only binding to the malate conducting TaALMT1 conformation. Our results demonstrate that picrotoxin inhibits the anion efflux mediated by TaALMT1 and could be used as a new inhibitor to distinguish between the anion transport and GABA transport modes for ALMT proteins. Further research to identify the functional roles of the residues that have been identified as important for picrotoxin binding is ongoing.

## 5. Conclusions

In conclusion, this study demonstrates that picrotoxin blocks the malate efflux mediated by *TaALMT1* in heterologous expression systems and in near-isogenic lines of wheat ET8 (Al-tolerant). In addition, picrotoxin has no effect on the GABA efflux mediated by ALMTs. It is likely that picrotoxin is an allosteric inhibitor of anion efflux, but further research is required to gain insights into its mechanism of action.

## Figures and Tables

**Figure 1 biology-11-01162-f001:**
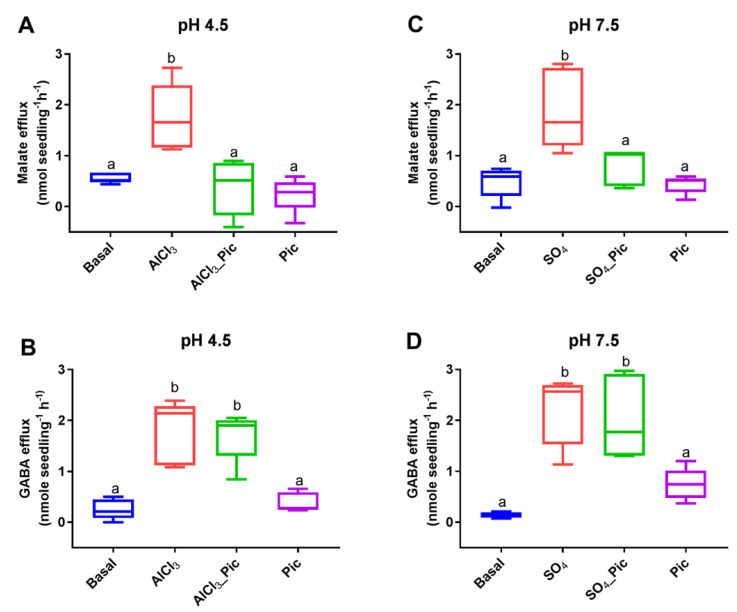
Inhibition of malate efflux but not GABA efflux from roots of Al-tolerant ET8 seedlings by picrotoxin. Aluminum (AlCl_3_; 100 μM) stimulated both malate and GABA efflux (**A**,**B**) at pH 4.5, but picrotoxin (Pic, 10 μM) only inhibited malate efflux. A corresponding experiment with anion activation (10 mM SO_4_^2−^) of malate efflux at pH 7.5 (**C**) showed the same inhibition of malate efflux but not GABA efflux (**D**). Data shown (*n* = 5) as box plots with letters representing significant differences based on one-way ANOVA.

**Figure 2 biology-11-01162-f002:**
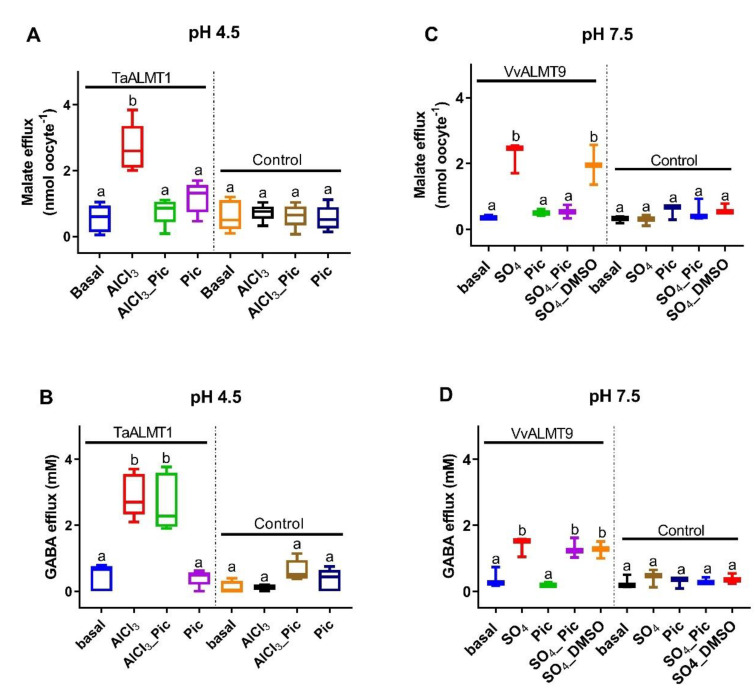
Malate efflux from *Xenopus* oocytes expressing ALMTs is blocked by picrotoxin (Pic, 10 µM) but not GABA efflux. (**A**,**B**) *TaALMT1* where malate and GABA efflux is activated by AlCl_3_ (100 µM). (**C**,**D**) *VvALMT9* is not activated by AlCl_3_ but can be activated by adding 10 mM Na_2_SO_4._ The dimethyl sulfoxide (DMSO) used to dissolve the picrotoxin had no effect on malate or GABA efflux. Controls are water-injected oocytes. Data shown for *TaALMT1* (*n* = 5) and *VvALMT9* (*n* = 3) are box plots, with different letters representing significant differences based on one-way ANOVA.

**Figure 3 biology-11-01162-f003:**
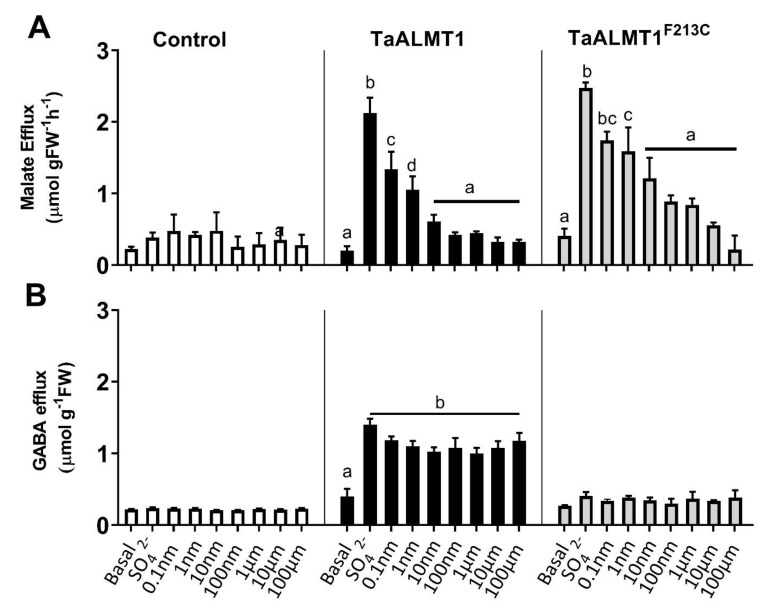
Inhibition of malate (**A**) efflux but not GABA efflux (**B**) from tobacco BY2 cells expressing *TaALMT1* or the mutant *TaALMT1*^F213C^ by picrotoxin. Malate and GABA efflux were activated by Na_2_SO_4_ (SO_4_^2−^ 10 mM; pH 7.5), and the effects of picrotoxin were tested in combination with Na_2_SO_4_ solution at concentrations ranging from 0.1 nM to 100 μM. Data are displayed as means ± SE (*n* = 3). Different letters represent significant differences between treatments using one-way ANOVA.

**Figure 4 biology-11-01162-f004:**
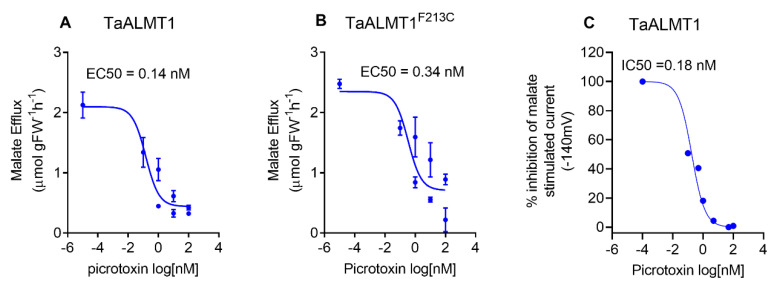
(**A**,**B**) Dose–response curves for picrotoxin inhibition of malate efflux (pH 7.5) from tobacco BY2 cells expressing *TaALMT1* or the GABA-insensitive mutant (*TaALMT1^F213C^*). (**C**) Inhibition of inward current (anion efflux) expressed as % inhibition from *Xenopus* oocytes expressing *TaALMT1*. Oocytes were preloaded with malate. Controls (water injected) were subtracted from the *TaALMT1* expressing oocytes. Dose response calculated from *n* = 3 for tobacco BY2 cells and *n* = 5 for *Xenopus* oocytes.

**Figure 5 biology-11-01162-f005:**
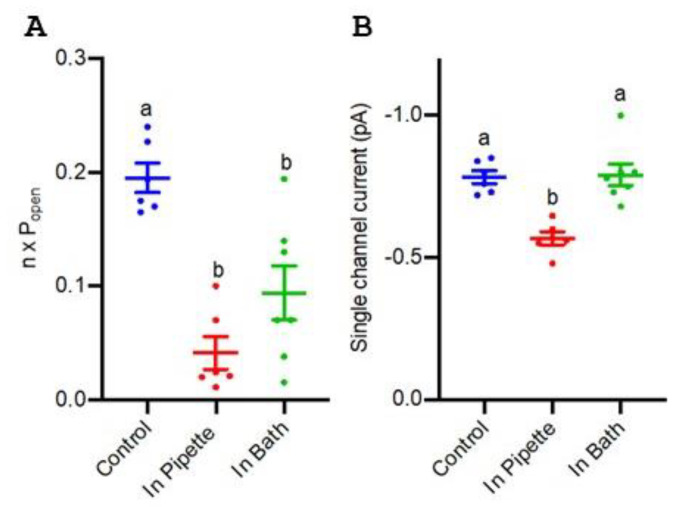
Summary of single-channel properties of *TaALMT1* channels recorded using the outside-out patch clamp technique on the plasma membrane of *TaALMT1-*expressing *Xenopus* oocytes. (**A**) Number of channels multiplied by the channel open probability (n x P_open_). (**B**) Single-channel amplitude at −160 mV with 35 mM: 25 mM malate (pipette:bath), pH = 7.2 (HEPES /Tris). Each point represents a single oocyte recording. Mean and SE are shown. Different letters represent significant differences between treatments using one-way ANOVA.

**Figure 6 biology-11-01162-f006:**
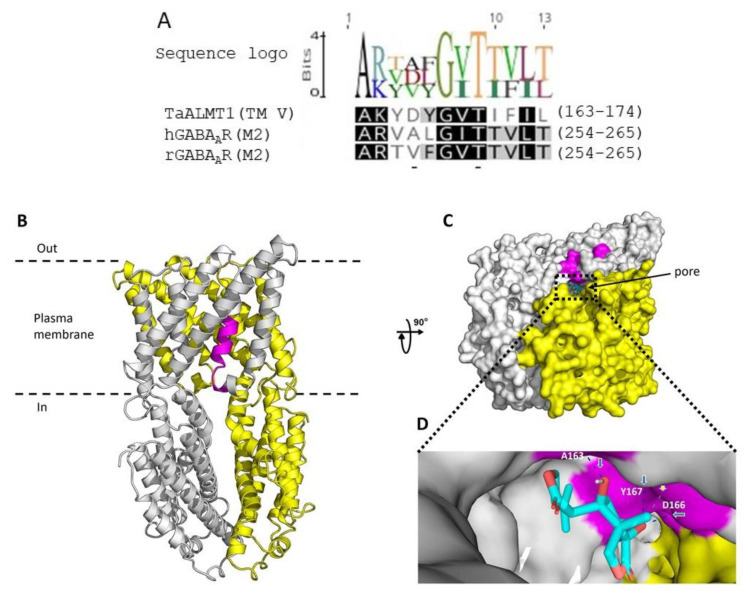
Possible binding site/s for picrotoxin in TaALMT1. (**A**) The GABA_A_ pore lining, TM2, of the α subunit of the mammalian GABA_A_ receptors has residues (valine and threonine) that are linked to picrotoxin binding. The TaALMT1 motif in TM5 has the residues aspartate and threonine (underlined) located in the same positions, and these residues may be involved in picrotoxin binding. The logo depicts the residue frequencies at each position within the motif in TaALMT1 and in mammalian GABA receptors. Identical residues are shaded in black; 80% similar residues are shaded in grey, and residues with similarity < 60% are unshaded. Full-sequence identifiers are TaALMT1 (DQ072260), the human GABA receptor alpha subunit (P14867), and the rat GABA receptor alpha subunit (P62813). (**B**) Homology model of TaALMT1 shown as ribbon representation viewed from the side of the plasma membrane. The TaALMT1 homodimer displays subunits A (grey) and B (yellow). The motif in TM5 is highlighted in magenta. (**C**) Picrotoxin docking to TaALMT1. Picrotoxin (cyan) docked into the pore of TaALMT1 and shown in a representative orientation (rotated 90° from B and viewed from the intracellular side). The dashed lines show the central ion pore. (**D**) Close-up of the putative picrotoxin binding site within TaALMT1 interacting with A163 (via H-bond), D166 (via H-bond and hydrophobic interactions), and Y167 (via H-bond). Blue arrows point to dashed blue lines representing H-bonds, whilst the white arrow points to dashed grey line indicating hydrophobic interactions. Figures created with PyMol (accessed on 10 April 2022; https://pymol.org/2/).

**Figure 7 biology-11-01162-f007:**
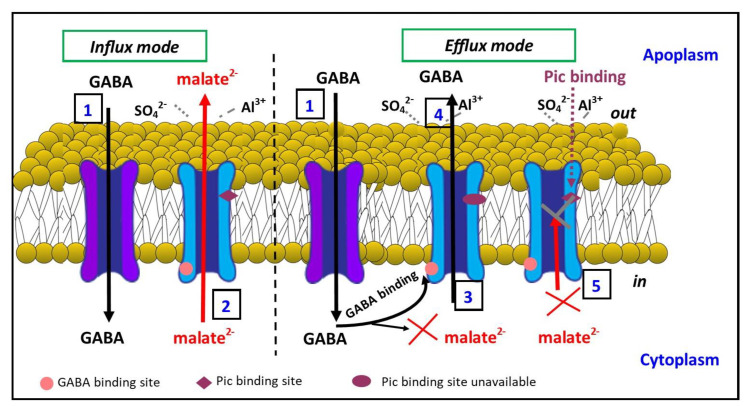
Schematic of GABA and malate transport through TaALMT1. In the influx mode, GABA transport can occur from the external side of the membrane via GABA transporters (**1**). In the presence of aluminium (Al^3+^) or transactivating anions such as sulphate (SO_4_^2−^), malate is effluxed (**2**) into the apoplasm across the plasma membrane via TaALMT1. GABA binding to the GABA binding site (**3**) on TaALMT1 is proposed to cause a conformational change that allows GABA efflux (**4**) but prevents malate efflux via TaALMT1. Picrotoxin (PTX) binding to the malate-transporting conformation of TaALMT1 blocks malate transport (**5**). GABA influx transporter (purple); malate/GABA efflux transporter TaALMT1 (blue).

## Data Availability

Not applicable.

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
