# Peer review of "Picrotoxin Delineates Different Transport Configurations for Malate and γ Aminobutyric Acid through TaALMT1"

_biology, 2022, doi:10.3390/biology11081162_

Round 1
Reviewer 1 Report
This paper clarifies the addition of picrotoxin affects the efflux of malate, but the efflux of GABA is not affected, which demonstrates that the efflux of malate and GABA may occur through the same pathway of ALMT proteins. There are some drawbacks that may be taken into account. The detailed comments are listed as follows:
1. Line2: delineates? This word is not closely related to the content described in the article.
2. Line11: The research background in the abstract is too much, and this part of the content is recommended to be moved to the introduction.
3. Line100: The materials used in the study should be described in detail.
4. Line221: The ordinate units of Figure 2A and 2C need to be unified.
5. Line229: Why are there no results and analysis for pH 4.5 in Tobacco BY2?
6. Line379: Does the state of charge affect the transport configurations of malate and GABA? This part needs further explanation.
Author Response
This paper clarifies the addition of picrotoxin affects the efflux of malate, but the efflux of GABA is not affected, which demonstrates that the efflux of malate and GABA may occur through the same pathway of ALMT proteins.
Dear Reviewer, Thank you for your comments. Please see below:
- Line2: delineates? This word is not closely related to the content described in the article.
Delineates refers “to describe, or set forth with accuracy or in detail” this word is used in the title to convey that picrotoxin “describes” different transport …..
- Line11: The research background in the abstract is too much, and this part of the content is recommended to be moved to the introduction.
The background is included to set the context for the research within, but on the reviewer’s suggestion, the abstract has been edited to make it more concise.
- Line100: The materials used in the study should be described in detail.
All methods have been described in detail in the respective sections of the methods.
- Line221: The ordinate units of Figure 2A and 2C need to be unified.
Thanks, this has been corrected now.
- Line229:Why are there no results and analysis for pH 4.5 in Tobacco BY2?
The experiment was also carried out at pH 4.5 and showed results similar to pH 7.5, so has not been included to avoid repetition. We have now added this information in to the manuscript as data not shown.
- Line379:Does the state of charge affect the transport configurations of malate and GABA? This part needs further explanation.
This is an active research area that is under further investigation, and we hope to have answers to this excellent question in the future.
Reviewer 2 Report
Difficult to understand article with general reading, very specific for the sector. Valid and varied but highly specific materials and methods. Good figures relating to the results, possibility of adding other examples of the pathways indicated in the introduction. Results well described. Not very innovative conclusions with studies to continue. The practical benefits of the study are unclear given the central role of GABA as a stress signalling in plants and its lack of inhibition by picrotoxin or other pharmaceutical agents.
Line 58-60 poorly expressed grammatically
Line 65 cryo-Em, specify the meaning
Line 171 2,4 d, specify the meaning
Line 319 was, because singular and not plural
Author Response
Dear Reviewer, Thank you for your comments. Please see below.
Line 58-60 poorly expressed grammatically
- This sentence has now been edited to make it clearer.
Line 65 cryo-Em, specify the meaning
- This has been changed to cryogenic electron microscopy.
Line 171 2,4 d, specify the meaning
- This information has now been added.
Line 319 was, because singular and not plural
- This has now been corrected to ”were”.
Reviewer 3 Report
This manuscript mainly reported that picrotoxin inhibited ALMT1-mediated anion efflux from wheat, but not associated with GABA transport, thus suggesting that picrotoxin may be a new inhibitory factor. Here are some of the problems in this article:
Abstract
Why could the author draw the conclusion of " picrotoxin…as a new inhibitor to study ion fluxes mediated by ALMT proteins that allow either GABA or anion transport" since the author has proved by experiments that the picrotoxin "has no effect on GABA transport"?
Introduction
Line 89-92: “…tolerant to aluminium at low pH by virtue of high expression of TaALMT1 [39] as well as an ALMT from Vitis vinifera (VvALMT9) expressed in different heterologous expression systems to assess if it inhibited anion and GABA efflux mediated by the ALMTs”, this sentence is confusing, please check it.
Materials and Methods
The authors point out that the concentration of GABA in plants is highly sensitive to stress and it plays a signal role in plants related to aluminum-activated ALMTs. So, here is a question, how does the author explicitly eliminate the interference of the “stress” (may be the stress of light, temperature, water, nutrition and other related subtle components)?
Results
3.1. Wheat root malate and GABA efflux
Line 207-208: In Figure1, it is clear that the treatment with the addition of toxin has a significant effect, and whereas the author mentioned that “Dimethyl sulfoxide (DMSO) used to dissolve picrotoxin had no effect on malate or GABA efflux (Figure S1)”, but you did not analyze the reason for this result which is what I am curious about.
Line 220-221: The sentence of “Dimethyl sulfoxide (DMSO) used to dissolve picrotoxin had no effect on malate or GABA efflux” is the same question. In addition, in Figure S1, the author did not add DMSO alone to get insignificant results. Another problem is that when the phrase of the abbreviation is defined, please use the abbreviation consistently in the subsequent expression of the article.
Discussion
It is suggested to add a part of the discussion that DMSO as a reagent to eliminate the effects of toxins has no effect on both malate and GABA efflux.
Author Response
Dear Reviewer, Thank you for your comments. Please see below.
Abstract
Why could the author draw the conclusion of " picrotoxin…as a new inhibitor to study ion fluxes mediated by ALMT proteins that allow either GABA or anion transport" since the author has proved by experiments that the picrotoxin "has no effect on GABA transport"?
The ALMT1 protein facilitates the transport of both malate and GABA in response to certain conditions. To better understand the regulation of malate flux and transport of GABA we need inhibitors that can block the flux of one while allowing the flux of the other. Thus, if we have inhibitors that can block malate while allowing GABA transport and vice versa, we can better understand the dual transport /flux though this protein and understand the role and importance of this transport in planta. This is an active area of research and we hope to have some exciting results soon.
Introduction
Line 89-92: “…tolerant to aluminium at low pH by virtue of high expression of TaALMT1 [39] as well as an ALMT from Vitis vinifera (VvALMT9) expressed in different heterologous expression systems to assess if it inhibited anion and GABA efflux mediated by the ALMTs”, this sentence is confusing, please check it.
This sentence has been edited to make it clearer.
Materials and Methods
The authors point out that the concentration of GABA in plants is highly sensitive to stress and it plays a signal role in plants related to aluminum-activated ALMTs. So, here is a question, how does the author explicitly eliminate the interference of the “stress” (may be the stress of light, temperature, water, nutrition and other related subtle components)?
This is a good question. GABA concentrations in the plants respond to diverse abiotic stresses. The ALMT proteins play a role in low pH soils where Al toxicity occurs, and our controlled experimental studies focus on this aspect. It would indeed be interesting to study plants grown in the field and investigate how GABA concentrations are affected by the above-mentioned stresses.
Results
Wheat root malate and GABA efflux
Line 207-208: In Figure1, it is clear that the treatment with the addition of toxin has a significant effect, and whereas the author mentioned that “Dimethyl sulfoxide (DMSO) used to dissolve picrotoxin had no effect on malate or GABA efflux (Figure S1)”, but you did not analyze the reason for this result which is what I am curious about.
- DMSO is used at very low concentrations (0.1%) and in the absence of its effect of fluxes, it is clear that Picrotoxin affects the fluxes and not DMSO which it is dissolved in.
Line 220-221: The sentence of “Dimethyl sulfoxide (DMSO) used to dissolve picrotoxin had no effect on malate or GABA efflux” is the same question. In addition, in Figure S1, the author did not add DMSO alone to get insignificant results. Another problem is that when the phrase of the abbreviation is defined, please use the abbreviation consistently in the subsequent expression of the article.
- DMSO was tested alone and as described above had no effect on fluxes from the oocytes. Abbreviation has been used now to replace the expansion.
Discussion
It is suggested to add a part of the discussion that DMSO as a reagent to eliminate the effects of toxins has no effect on both malate and GABA efflux.
- Thanks this has now been added to the discussion.
Round 2
Reviewer 3 Report
We have read the revised manuscripts and response made by the author, and each problem has been explained in general. Thus, it has met the standards which could be published in Biology.